# Biclustering Using Message Passing

**Luke O'Connor**
Bioinformatics and Integrative Genomics
Harvard University
Cambridge, MA 02138
loconnor@g.harvard.edu

**Soheil Feizi**
Electrical Engineering and Computer Science
Massachusetts Institute of Technology
Cambridge, MA 02139
sfeizi@mit.edu

## Abstract

Biclustering is the analog of clustering on a bipartite graph. Existent methods infer biclusters through local search strategies that find one cluster at a time; a common technique is to update the row memberships based on the current column memberships, and vice versa. We propose a biclustering algorithm that maximizes a global objective function using message passing. Our objective function closely approximates a general likelihood function, separating a cluster size penalty term into row- and column-count penalties. Because we use a global optimization framework, our approach excels at resolving the overlaps between biclusters, which are important features of biclusters in practice. Moreover, Expectation-Maximization can be used to learn the model parameters if they are unknown. In simulations, we find that our method outperforms two of the best existing biclustering algorithms, ISA and LAS, when the planted clusters overlap. Applied to three gene expression datasets, our method finds coregulated gene clusters that have high quality in terms of cluster size and density.

## 1 Introduction

The term *biclustering* has been used to describe several distinct problems variants. In this paper, In this paper, we consider the problem of biclustering as a bipartite analogue of clustering: Given an $N \times M$ matrix, a bicluster is a subset of rows that are heavily connected to a subset of columns. In this framework, biclustering methods are data mining techniques allowing simultaneous clustering of the rows and columns of a matrix. We suppose there are two possible distributions for edge weights in the bipartite graph: a within-cluster distribution and a background distribution. Unlike in the traditional clustering problem, in our setup, biclusters may overlap, and a node may not belong to any cluster. We emphasize the distinction between biclustering and the bipartite analog of graph partitioning, which might be called *bipartitioning*.

Biclustering has several noteworthy applications. It has been used to find modules of coregulated genes using microarray gene expression data [1] and to predict tumor phenotypes from their genotypes [2]. It has been used for document classification, clustering both documents and related words simultaneously [3]. In all of these applications, biclusters are expected to overlap with each other, and these overlaps themselves are often of interest (e.g., if one wishes to explore the relationships between document topics).

The biclustering problem is NP-hard (see Proposition 1). However, owing to its practical importance, several heuristic methods using local search strategies have been developed. A popular approach is to search for one bicluster at a time by iteratively assigning rows to a bicluster based on the columns, and vice versa. Two algorithms based on this approach are ISA [4] and LAS [5]. Another approach is an exhaustive search for complete bicliques used by Bimax [6]. This approach fragments large noisy clusters into small complete ones. SAMBA [7] uses a heuristic combinatorial search for locally optimal biclusters, motivated by an exhaustive search algorithm that is exponential

in the maximum degree of the nodes. For more details about existent biclustering algorithms, and performance comparisons, see references [6] and [8]. Existent biclustering methods have two major shortcomings: first, they apply a local optimality criterion to each bicluster individually. Because a collection of locally optimal biclusters might not be globally optimal, these local methods struggle to resolve overlapping clusters, which arise frequently in many applications. Second, the lack of a well-defined global objective function precludes an analytical characterization of their expected results.

Global optimization methods have been developed for problems closely related to biclustering, including clustering. Unlike most biclustering problem formulations, these are mostly partitioning problems: each node is assigned to one cluster or category. Major recent progress has been made in the development of spectral clustering methods (see references [9] and [10]) and message-passing algorithms (see [11], [12] and [13]). In particular, Affinity Propagation [12] maximizes the sum of similarities to one central exemplar instead of overall cluster density. Reference [14] uses variational expectation-maximization to fit the latent block model, which is a binary model in which each row or column is assigned to a row or column cluster, and the probability of an edge is dictated by the respective cluster memberships. Row and column clusters that are not paired to form biclusters.

In this paper, we propose a message-passing algorithm that searches for a globally optimal collection of possibly overlapping biclusters. Our method maximizes a likelihood function using an approximation that separates a cluster-size penalty term into a row-count penalty and a column-count penalty. This decoupling enables the messages of the max-sum algorithm to be computed efficiently, effectively breaking an intractable optimization into a pair of tractable ones that can be solved in nearly linear time. When the underlying model parameters are unknown, they can be learned using an expectation-maximization approach.

Our approach has several advantages over existing biclustering algorithms: the objective function of our biclustering method has the flexibility to handle diverse statistical models; the max-sum algorithm is a more robust optimization strategy than commonly used iterative approaches; and in particular, our global optimization technique excels at resolving overlapping biclusters. In simulations, our method outperforms two of the best existing biclustering algorithms, ISA and LAS, when the planted clusters overlap. Applied to three gene expression datasets, our method found biclusters of high quality in terms of cluster size and density.

## 2 Methods

### 2.1 Problem statement

Let $G = (V, W, E)$ be a weighted bipartite graph, with vertices $V = (1, ..., N)$ and $W = (1, ..., M)$, connected by edges with non-negative weights: $E : V \times W \to [0, \infty)$. Let $V_1, ..., V_K \subset V$ and $W_1, ..., W_K \subset W$. Let $(V_k, W_k) = \{(i, j) : i \in V_k, j \in W_k\}$ be a bicluster: Graph edge weights $e_{ij}$ are drawn independently from either a *within-cluster distribution* or a *background distribution* depending on whether, for some $k$, $i \in V_k$ and $j \in W_k$. In this paper, we assume that the within-cluster and background distributions are homogenous. However, our formulation can be extended to a general case in which the distributions are row- or column-dependent.

Let $c_{ij}^k$ be the indicator for $i \in V_k$ and $j \in W_k$. Let $c_{ij} \triangleq \min(1, \sum_k c_{ij}^k)$ and let $\mathbf{c} \triangleq (c_{ij}^k)$.

**Definition 1** (Biclustering Problem). *Let $G = (V, W, E)$ be a bipartite graph with biclusters $(V_1, W_1), ..., (V_K, W_K)$, within-cluster distribution $f_1$ and background distribution $f_0$. The problem is to find the maximum likelihood cluster assignments (up to reordering):*

$$\hat{\mathbf{c}} = \arg\max_{\mathbf{c}} \sum_{(i,j)} c_{ij} \log \frac{f_1(e_{ij})}{f_0(e_{ij})}, \tag{1}$$

$$c_{ij}^k = c_{rs}^k = 1 \Rightarrow c_{is}^k = c_{rj}^k = 1, \quad \forall i, r \in V, \forall j, s \in W.$$

Figure 1 demonstrates the problem qualitatively for an unweighted bipartite graph. In general, the combinatorial nature of a biclustering problem makes it computationally challenging.

**Proposition 1.** *The clique problem can be reduced to the maximum likelihood problem of Definition (1). Thus, the biclustering problem is NP-hard.*

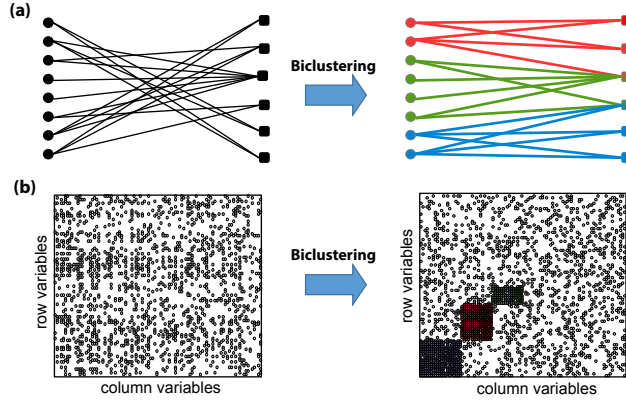

Figure 1: Biclustering is the analogue of clustering on a bipartite graph. (a) Biclustering allows nodes to be reordered in a manner that reveals modular structures in the bipartite graph. (b) The rows and columns of an adjacency matrix are similarly biclustered and reordered.

*Proof.* Proof is provided in Supplementary Note 1. □

## 2.2   BCMP objective function

In this section, we introduce the global objective function considered in the proposed biclustering algorithm called *Biclustering using Message Passing* (BCMP). This objective function approximates the likelihood function of Definition 1. Let $l_{ij} = \log \frac{f_1(e_{ij})}{f_0(e_{ij})}$ be the log-likelihood ratio score of tuple $(i, j)$. Thus, the likelihood function of Definition 1 can be written as $\sum c_{ij} l_{ij}$. If there were no consistency constraints in the Optimization (1), an optimal maximum likelihood biclustering solution would be to set $c_{ij} = 1$ for all tuples with positive $l_{ij}$. Our key idea is to enforce the consistency constraints by introducing a cluster-size penalty function and shifting the log-likelihood ratios $l_{ij}$ to recoup this penalty. Let $N_k$ and $M_k$ be the number of rows and columns, respectively, assigned to cluster $k$. We have,

$$
\begin{aligned}
\sum_{(i,j)} c_{ij} l_{ij} &\overset{(a)}{\approx} \sum_{(i,j)} c_{ij} \max(0, l_{ij} + \delta) - \delta \sum_{(i,j)} c_{ij} \\
&\overset{(b)}{=} \sum_{(i,j)} c_{ij} \max(0, l_{ij} + \delta) + \delta \sum_{(i,j)} \max(0, -1 + \sum_k c_{ij}^k) - \delta \sum_k N_k M_k \\
&\overset{(c)}{\approx} \sum_{(i,j)} c_{ij} \max(0, l_{ij} + \delta) + \delta \sum_{(i,j)} \max(0, -1 + \sum_k c_{ij}^k) - \frac{\delta}{2} \sum_k r_k N_k^2 + r_k^{-1} M_k^2.
\end{aligned}
\tag{2}
$$

The approximation $(a)$ holds when $\delta$ is large enough that thresholding $l_{ij}$ at $-\delta$ has little effect on the resulting objective function. In equation $(b)$, we have expressed the second term of $(a)$ in terms of a cluster size penalty $-\delta N_k M_k$, and we have added back a term corresponding to the overlap between clusters. Because a cluster-size penalty function of the form $N_k M_k$ leads to an intractable optimization in the max-sum framework, we approximate it using a decoupling approximation $(c)$ where $r_k$ is a cluster shape parameter:

$$
2N_k M_k \approx r_k N_k^2 + r_k^{-1} M_k^2,
\tag{3}
$$

when $r_k \approx M_k / N_k$. The cluster-shape parameter can be iteratively tuned to fit the estimated biclusters.

Following equation (2), the BCMP objective function can be separated into three terms as follows:

$$F(\mathbf{c}) = \sum_{i,j} \tau_{ij} + \sum_k \eta_k + \sum_k \mu_k, \tag{4}$$

$$\begin{cases} \tau_{ij} = \ell_{ij} \min(1, \sum_k c_{ij}^k) + \delta \max(0, \sum_k c_{ij}^k - 1) & \forall (i,j) \in V \times W, \\ \eta_k = -\frac{\delta}{2} r_k N_k^2 & \forall 1 \le k \le K, \\ \mu_k = -\frac{\delta}{2} r_k^{-1} M_k^2 & \forall 1 \le k \le K \end{cases} \tag{5}$$

Here $\tau_{ij}$, the tuple function, encourages heavier edges of the bipartite graph to be clustered. Its second term compensates for the fact that when biclusters overlap, the cluster-size penalty functions double-count the overlapping regions. $\ell_{ij} \triangleq \max(0, l_{ij} - \delta)$ is the shifted log-likelihood ratio for observed edge weight $e_{ij}$. $\eta_k$ and $\mu_k$ penalize the number of rows and columns of cluster $k$, $N_k$ and $M_k$, respectively. Note that by introducing a penalty for each nonempty cluster, the number of clusters can be learned, and finding weak, spurious clusters can be avoided (see Supplementary Note 3.3).

Now, we analyze BCMP over the following model for a binary or unweighted bipartite graph:

**Definition 2.** *The binary biclustering model is a generative model for $N \times M$ bipartite graph $(V, W, E)$ with $K$ biclusters distributed by uniform sampling with replacement, allowing for overlapping clusters. Within a bicluster, edges are drawn independently with probability $p$, and outside of a bicluster, they are drawn independently with probability $q < p$.*

In the following, we assume that $p, q$, and $K$ are given. We discuss the case that the model parameters are unknown in Section 2.4. The following proposition shows that optimizing the BCMP objective function solves the problem of Definition 1 in the case of the binary model:

**Proposition 2.** *Let $(e_{ij})$ be a matrix generated by the binary model described in Definition 2. Suppose $p, q$ and $K$ are given. Suppose the maximum likelihood assignment of edges to biclusters, $\arg\max(P(data|\mathbf{c}))$, is unique up to reordering. Let $r_k = M_k'/N_k'$ be the cluster shape ratio for the $k$-th maximum likelihood cluster. Then, by using these values of $r_k$, setting $\ell_{ij} = e_{ij}$, for all $(i,j)$, with cluster size penalty*

$$\frac{\delta}{2} = -\frac{\log(\frac{1-p}{1-q})}{2\log(\frac{p(1-q)}{q(1-p)})}, \tag{6}$$

*we have,*

$$\arg\max_{\mathbf{c}}(P(data|\mathbf{c})) = \arg\max_{\mathbf{c}}(F(\mathbf{c})). \tag{7}$$

*Proof.* The proof follows the derivation of equation (2). It is presented in Supplementary Note 2. □

**Remark 1.** *In the special case when $q = 1 - p \in (0, 1/2)$, according to equation (6), we have $\frac{\delta}{2} = 1/4$. This is suggested as a reasonable initial value to choose when the true values of $p$ and $q$ are unknown; see Section 2.4 for a discussion of learning the model parameters.*

The assumption that $r_k = N_k'/M_k'$ may seem rather strong. However, it is essential as it justifies the decoupling equation (3) that enables a linear-time algorithm. In practice, if the initial choice of $r_k$ is close enough to the actual ratio that a cluster is detected corresponding to the real cluster, $r_k$ can be tuned to find the true value by iteratively updating it to fit the estimated bicluster. This iterative strategy works well in our simulations. For more details about automatically tuning the parameter $r_k$, see Supplementary Note 3.1.

In a more general statistical setting, log-likelihood ratios $l_{ij}$ may be unbounded below, and the first step $(a)$ of derivation (2) is an approximation; setting $\delta$ arbitrarily large will eventually lead to instability in the message updates.

## 2.3 Biclustering Using Message Passing

In this section, we use the max-sum algorithm to optimize the objective function of equation (4). For a review of the max-sum message update rules, see Supplementary Note 4. There are $NM$ function nodes for the functions $\tau_{ij}$, $K$ function nodes for the functions $\eta_k$, and $K$ function nodes for the functions $\mu_k$. There are $NMK$ binary variables, each attached to three function nodes: $c_{ij}^k$ is attached to $\tau_{ij}, \eta_k$, and $\mu_k$ (see Supplementary Figure 1). The incoming messages from these function nodes are named $t_{ij}^k$, $n_{ij}^k$, and $m_{ij}^k$, respectively. In the following, we describe messages for $c_{ij}^k = c_{12}^1$; other messages can be computed similarly.

First, we compute $t_{12}^1$:

$$t_{12}^1(x) \overset{(a)}{=} \max_{c_{12}^2,\ldots,c_{12}^K} \left[\tau_{12}(x, c_{12}^2, \ldots, c_{12}^K) + \sum_{k\neq 1} m_{12}^k(c_{12}^k) + n_{12}^k(c_{12}^k)\right] \tag{8}$$

$$\overset{(b)}{=} \max_{c_{12}^2,\ldots,c_{12}^K} \left[\ell_{12}\min(1, \sum_k c_{12}^k) + \delta\max(0, \sum_k c_{12}^k - 1) + \sum_{k\neq 1} c_{12}^k(m_{12}^k + n_{12}^k)\right] + d_1$$

where $d_1 = \sum_{k\neq 1} m_{12}^k(0) + n_{12}^k(0)$ is a constant. Equality (a) comes from the definition of messages according to equation (6) in the Supplement. Equality (b) uses the definition of $\tau_{12}$ of equation (5) and the definition of the scalar message of equation (8) in the Supplement. We can further simplify $t_{12}$ as follows:

$$\begin{cases} t_{12}^1(1) - d_1 \overset{(c)}{=} \ell_{12} + \sum_{k\neq 1} max(0, \delta + m_{12}^k + n_{12}^k), \\ t_{12}^1(0) - d_1 \overset{(d)}{=} \ell_{12} - \delta + \sum_{k\neq 1} max(0, \delta + m_{12}^k + n_{12}^k), & \text{if } \exists k, n_{12}^k + m_{12}^k + \delta > 0, \\ t_{12}^1(0) - d_1 \overset{(e)}{=} max(0, \ell_{12} + max_{k\neq 1}(m_{12}^k + n_{12}^k)), & \text{otherwise .} \end{cases} \tag{9}$$

If $c_{12}^1 = 1$, we have $\min(1, \sum_k c_{12}^k) = 1$, and $\max(0, \sum_k c_{12}^k - 1) = \sum_{k\neq 1} c_{12}^k$. These lead to equality (c). A similar argument can be made if $c_{12}^1 = 0$ but there exists a $k$ such that $n_{12}^k + m_{12}^k + \delta > 0$. This leads to equality (d). If $c_{12}^1 = 0$ and there is no $k$ such that $n_{12}^k + m_{12}^k + \delta > 0$, we compare the increase obtained by letting $c_{12}^k = 1$ (i.e., $\ell_{12}$) with the penalty (i.e., $m_{12}^k + n_{12}^k$), for the best $k$. This leads to equality (e).

**Remark 2.** *Computation of $t_{ij}^1, \ldots, t_{ij}^k$ using equality (d) costs $\mathcal{O}(K)$, and not $\mathcal{O}(K^2)$, as the summation need only be computed once.*

Messages $m_{12}^1$ and $n_{12}^1$ are computed as follows:

$$\begin{cases} m_{12}^1(x) = \max_{\mathbf{c}^1|c_{12}^1=x} \left[\mu_1(\mathbf{c}^1) + \sum_{(i,j)\neq(1,2)} t_{ij}^1(c_{ij}^1) + n_{ij}^1(c_{ij}^1)\right], \\ n_{12}^1(x) = \max_{\mathbf{c}^1|c_{12}^1=x} \left[\eta_1(\mathbf{c}^1) + \sum_{(i,j)\neq(1,2)} t_{ij}^1(c_{ij}^1) + m_{ij}^1(c_{ij}^1)\right], \end{cases} \tag{10}$$

where $\mathbf{c}^1 = \{c_{ij}^1 : i \in V, j \in W\}$. To compute $n_{12}^1$ in constant time, we perform a preliminary optimization, ignoring the effect of edge $(1, 2)$:

$$\arg\max_{\mathbf{c}^1} -\frac{\delta}{2}N_1^2 + \sum_{(i,j)} t_{ij}^1(c_{ij}^1) + m_{ij}^1(c_{ij}^1). \tag{11}$$

Let $s_i = \sum_{j=1}^M \max(0, m_{ij}^1 + t_{ij}^1)$ be the sum of positive incoming messages of row $i$. The function $\eta_1$ penalizes the number of rows containing some nonzero $c_{ij}^1$: if any message along that row is included, there is no additional penalty for including every positive message along that row. Thus, optimization (11) is computed by deciding which rows to include. This can be done efficiently through sorting: we sort row sums $s_{(1)}, \ldots, s_{(N)}$ at a cost of $O(N \log N)$. Then we proceed from largest to smallest, including row $(N + 1 - i)$ if the marginal penalty $\frac{\delta}{2}(i^2 - (i-1)^2) = \frac{\delta}{2}(2i - 1)$ is less than $s_{(N+1-i)}$. After solving optimization (11), the messages $n_{12}^1, \ldots, n_{N2}^1$ can be computed in linear time, as we explain in Supplementary Note 5.

**Remark 3.** *Computation of $n_{ij}^k$ through sorting costs $\mathcal{O}(N \log N)$.*

**Proposition 3** (Computational Complexity of BCMP). *The computational complexity of BCMP over a bipartite graph with $N$ rows, $M$ columns, and $K$ clusters is $\mathcal{O}(K(N + \log M)(M + \log N))$.*

*Proof.* For each iteration, there are $NM$ messages $t_{ij}$ to be computed at cost $\mathcal{O}(K)$ each. Before computing $(n_{ij}^k)$, there are $K$ sorting steps at a cost of $\mathcal{O}(M \log M)$, after which each message may be computed in constant time. Likewise, there are $K$ sorting steps at a cost of $\mathcal{O}(N \log N)$ each before computing $(m_{ij}^k)$. □

We provide an empirical runtime example of the algorithm in Supplementary Figure 3.

### 2.4  Parameter learning using Expectation-Maximization

In the BCMP objective function described in Section 2.2, the parameters of the generative model were used to compute the log-likelihood ratios $(l_{ij})$. In practice, however, these parameters may be unknown. Expectation-Maximization (EM) can be used to estimate these parameters. The use of EM in this setting is slightly unorthodox, as we estimate the hidden labels (cluster assignments) in the M step instead of the E step. However, the distinction between parameters and labels is not intrinsic in the definition of EM [15] and the true ML solution is still guaranteed to be a fixed point of the iterative process. Note that it is possible that the EM iterative procedure leads to a locally optimal solution and therefore it is recommended to use several random re-initializations for the method.

The EM algorithm has three steps:

- **Initialization**: We choose initial values for the underlying model parameters $\theta$ and compute the log-likelihood ratios $(l_{ij})$ based on these values, denoting by $F_0$ the initial objective function.
- **M step**: We run BCMP to maximize the objective $F_i(\mathbf{c})$. We denote the estimated cluster assignments by by $\hat{\mathbf{c}}_i$ .
- **E step**: We compute the expected-log-likelihood function as follows:

$$F_{i+1}(\mathbf{c}) = E_\theta[\log P((e_{ij})|\theta)|\mathbf{c} = \hat{\mathbf{c}}_i] = \sum_{(i,j)} E_\theta[\log P(e_{ij}|\theta)|\mathbf{c} = \hat{\mathbf{c}}_i]. \quad (12)$$

Conveniently, the expected-likelihood function takes the same form as the original likelihood function, with an input matrix of *expected* log-likelihood ratios. These can be computed efficiently if conjugate priors are available for the parameters. Therefore, BCMP can be used to maximize $F_{i+1}$. The algorithm terminates upon failure to improve the estimated likelihood $F_i(\hat{\mathbf{c}}_i)$.

For a discussion of the application of EM to the binary and Gaussian models, see Supplementary Note 6. In the case of the binary model, we use uniform Beta distributions as conjugate priors for $p$ and $q$, and in the case of the Gaussian model, we use inverse-gamma-normal distributions as the priors for the variances and means. Even when convenient priors are not available, EM is still tractable as long as one can sample from the posterior distributions.

## 3  Evaluation results

We compared the performance of our biclustering algorithm with two methods, ISA and LAS, in simulations and in real gene expression datasets (Supplementary Note 8). ISA was chosen because it performed well in comparison studies [6] [8], and LAS was chosen because it outperformed ISA in preliminary simulations. Both ISA and LAS search for biclusters using iterative refinement. ISA assigns rows iteratively to clusters fractionally in proportion to the sum of their entries over columns. It repeats the same for column-cluster assignments, and this process is iterated until convergence. LAS uses a similar greedy iterative search without fractional memberships, and it masks already-detected clusters by mean subtraction.

In our simulations, we generate simulated bipartite graphs of size 100x100. We planted (possibly overlapping) biclusters as full blocks with two noise models:

- *Bernoulli noise*: we drew edges according to the binary model of Definition 2 with varying noise level $q = 1 - p$.

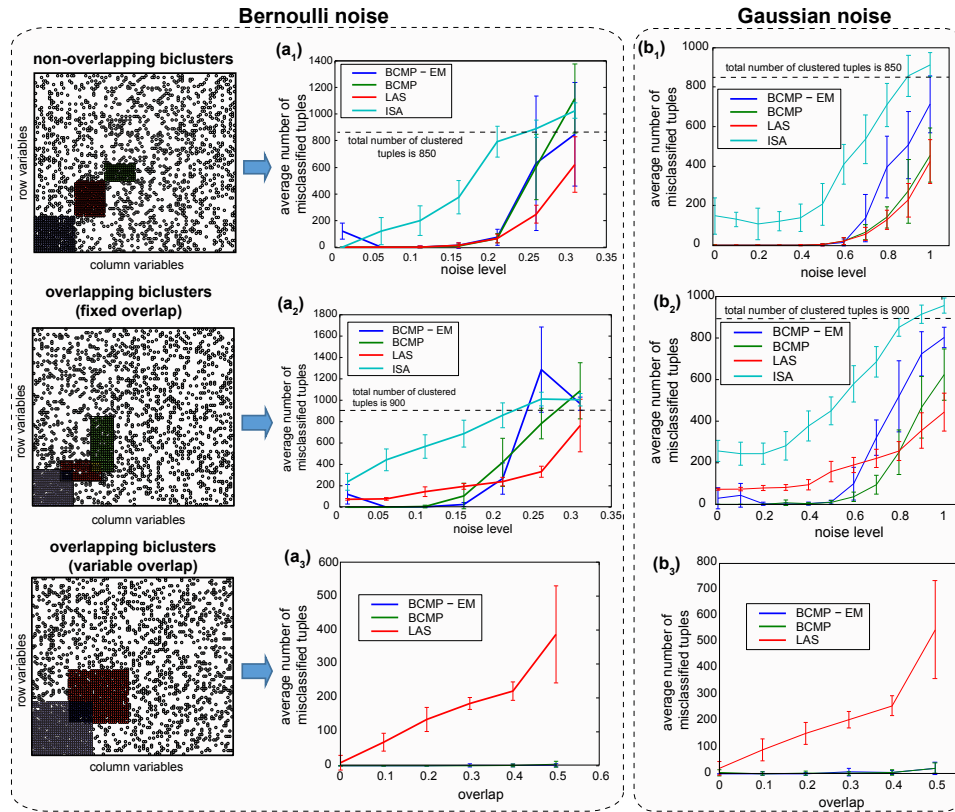

Figure 2: Performance comparison of the proposed method (BCMP) with ISA and LAS, for Bernoulli and Gaussian models, and for overlapping and non-overlapping biclusters. On the $y$ axis is the total number of misclassified row-column pairs. Either the noise level or the amount of overlap is on the $x$ axis.

- *Gaussian noise*: we drew edge weights within and outside of biclusters from normal distributions $N(1, \sigma^2)$ and $N(0, \sigma^2)$, respectively, for different values of $\sigma$.

For each of these cases, we ran simulations on three setups (see Figure 2):

- *Non-overlapping clusters:* three non-overlapping biclusters were planted in a $100 \times 100$ matrix with sizes $20 \times 20$, $15 \times 20$, and $15 \times 10$. We varied the noise level.

- *Overlapping clusters with fixed overlap:* Three overlapping biclusters with fixed overlaps were planted in a $100 \times 100$ matrix with sizes $20 \times 20$, $20 \times 10$, and $10 \times 30$. We varied the noise level.

- *Overlapping clusters with variable overlap:* we planted two $30 \times 30$ biclusters in a $100 \times 100$ matrix with variable amount of overlap between them, where the amount of overlap is defined as the fraction of rows and columns shared between the two clusters. We used Bernoulli noise level $q = 1 - p = 0.15$, and Gaussian noise level $\sigma = 0.7$.

The methods used have some parameters to set. Pseudocode for BCMP is presented in Supplementary Note 10. Here are the parameters that we used to run each method:

- *BCMP method with underlying parameters given*: We computed the input matrix of shifted log-likelihood ratios following the discussion in Section 2.2. The number of biclusters $K$ was given. We initialized the cluster-shape parameters $r_k$ at 1 and updated them as discussed in Supplementary Note 3.1. In the case of Bernoulli noise, following Proposition 2 and Remark 1, we set $\ell_{ij} = e_{ij}$ and $\frac{\delta}{2} = 1/4$. In the case of Gaussian noise, we chose a threshold $\delta$ to maximize the unthresholded likelihood (see Supplementary Note 3.2).

- *BCMP - EM method:* Instead of taking the underlying model parameters as given, we estimated them using the procedure described in Section 2.4 and Supplementary Note 6.

We used identical, uninformative priors on the parameters of the within-cluster and null distributions.

- *ISA method*: We used the same threshold ranges for both rows and columns, attempting to find best-performing threshold values for each noise level. These values were mostly around $1.5$ for both noise types and for all three dataset types. We found positive biclusters, and used 20 reinitializations. Out of these 20 runs, we selected the best-performing run.

- *LAS method*: There were no parameters to set. Since $K$ was given, we selected the first $K$ biclusters discovered by LAS, which marginally increased its performance.

Evaluation results of both noise models and non-overlapping and overlapping biclusters are shown in Figure 2. In the non-overlapping case, BCMP and LAS performed similarly well, better than ISA. Both of these methods made few or no errors up until noise levels $q = 0.2$ and $\sigma = .6$ in Bernoulli and Gaussian cases, respectively. When the parameters had to be estimated using EM, BCMP performed worse for higher levels of Gaussian noise but well otherwise. ISA outperformed BCMP and LAS at very high levels of Bernoulli noise; at such a high noise level, however, the results of all three algorithms are comparable to a random guess.

In the presence of overlap between biclusters, BCMP outperformed both ISA and LAS except at very high noise levels. Whereas LAS and ISA struggled to resolve these clusters even in the absence of noise, BCMP made few or no errors up until noise levels $q = 0.2$ and $\sigma = .6$ in Bernoulli and Gaussian cases, respectively. Notably, the overlapping clusters were more asymmetrical, demonstrating the robustness of the strategy of iteratively tuning $r_k$ in our method. In simulations with variable overlaps between biclusters, for both noise models, BCMP outperformed LAS significantly, while the results for the ISA method were very poor (data not shown). These results demonstrate that BCMP excels at inferring overlapping biclusters.

## 4 Discussion and future directions

In this paper, we have proposed a new biclustering technique called *Biclustering Using Message Passing* that, unlike existent methods, infers a globally optimal collection of biclusters rather than a collection of locally optimal ones. This distinction is especially relevant in the presence of overlapping clusters, which are common in most applications. Such overlaps can be of importance if one is interested in the relationships among biclusters. We showed through simulations that our proposed method outperforms two popular existent methods, ISA and LAS, in both Bernoulli and Gaussian noise models, when the planted biclusters were overlapping. We also found that BCMP performed well when applied to gene expression datasets.

Biclustering is a problem that arises naturally in many applications. Often, a natural statistical model for the data is available; for example, a Poisson model can be used for document classification (see Supplementary Note 9). Even when no such statistical model will be available, BCMP can be used to maximize a heuristic objective function such as the *modularity function* [17]. This heuristic is preferable to clustering the original adjacency matrix when the degrees of the nodes vary widely; see Supplementary Note 7.

The same optimization strategy used in this paper for biclustering can also be applied to perform clustering, generalizing the graph-partitioning problem by allowing nodes to be in zero or several clusters. We believe that the flexibility of our framework to fit various statistical and heuristic models will allow BCMP to be used in diverse clustering and biclustering applications.

## Acknowledgments

We would like to thank Professor Manolis Kellis and Professor Muriel Médard for their advice and support. We would like to thank the Harvard Division of Medical Sciences for supporting this project.

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
