[Supplementary Material]

Supplementary Notes
# Biclustering Using Message Passing

**Luke O'Connor**
Bioinformatics and Integrative Genomics
Harvard University
Cambridge, MA 02138
loconnor@g.harvard.edu

**Soheil Feizi**
Electrical Engineering and Computer Science
Massachusetts Institute of Technology
Cambridge, MA 02139
sfeizi@mit.edu

## 1 Proof of Proposition 1

In the case $K = 1$ the problem (1) is equivalent to the problem of finding a maximum weight biclique on the graph $G = (U, V, (l_{ij}))$ with edge weights $l_{ij} = \log \frac{f_1(e_{ij})}{f_0(e_{ij})}$. This problem contains as a special case the problem of finding a maximum size complete clique on a graph $(V, E)$: let $U = V$ and $l_{ii} = 1$. For $i \neq j$, let $l_{ij} = l_{ji} = 0$ if $(i, j) \in E$ and $-\infty$ otherwise. Suppose the maximal clique on $(V, E)$ is $W$. Then the biclique $(W, W)$ has weight $|W|$. For every other biclique $(U', V')$, if its weight is not $-\infty$, then $U' \cap V'$ must correspond to a complete clique on $(V, E)$; its weight is $|U' \cap V'| \leq |W|$, thus $W$ is a solution to (1) on $G$, and the problem (1) is NP-hard.

## 2 Proof of Proposition 2

This proof is essentially a pedantic version of the derivation of equation (2) in the main text. Given a putative set of biclusters $(V_1, W_1), ..., (V_K, W_k)$, let $\mathbf{c} = (c_{ij})$ be the $N \times M$ matrix with ones where $i, j$ are in the same bicluster and zeroes elsewhere. If the data is $E = (e_{ij})$, with ones where edges are observed, then the probability of the data given these biclusters is just the product of the probabilities of the tuples,

$$P(E|I) = \Pi_{i \leq N, j \leq M} t_{ij}$$

where,

$$t_{ij} = \begin{cases} p & e_{ij} = c_{ij} = 1 \\ q & e_{ij} = 1, c_{ij} = 0 \\ 1 - p & e_{ij} = 0, c_{ij} = 1 \\ 1 - q & e_{ij} = c_{ij} = 0 \end{cases} \tag{1}$$

The maximum likelihood set of biclusters maximize this expression. Equivalently, we maximize the sum, over all tuples $(i, j)$ placed in a bicluster, of the log-likelihood ratios, $\log(p/q)$ if $e_{ij} = 1$ and $\log((1-p)/(1-q))$ otherwise.

Let the input matrix be $(\ell_{ij}) = (e_{ij})$, and let the cluster-size penalty coefficient be

$$\frac{\delta}{2} = -\frac{\log(\frac{1-p}{1-q})}{2 \log(\frac{p(1-q)}{q(1-p)})}. \tag{2}$$

Let $c_{ij}$ be the indicator for row-column pair $ij$ being in cluster $k$ for some $k$. First,

$$F(\mathbf{c}) = \sum_{ij} c_{ij}(\ell_{ij} + (\sum_k c_{ij}^k - 1) \cdot \delta) - \frac{\delta}{2} \sum_k r_k N_k^2 + r_k^{-1} M_k^2$$

$$= \sum c_{ij}(\ell_{ij} + (\sum_k c_{ij}^k - 1) \cdot \delta) - \delta[\sum_k N_k M_k + \frac{1}{2}(r_k N_k - r_k^{-1} M_k)^2].$$

Now, if each $r_k = M_k/N_k$, then the terms $\frac{\delta}{2}(rN_k - r^{-1}M_k)^2$ drop out; otherwise, there is an additional penalty. The additional penalty cannot hurt, since it is zero when $\mathbf{c} = \mathbf{c}'$ and nonpositive otherwise. Notice that if a tuple is assigned to more than one cluster, the term $(\sum_k c_{ij}^k - 1)\frac{\delta}{2}$ cancels the fact that it is counted in two cluster-size penalty functions. By dropping the cluster shape term we get

$$F(\mathbf{c}) \geq \sum c_{ij}(\ell_{ij} + (\sum_k c_{ij}^k - 1) \cdot \delta) - \delta \sum_k N_k M_k$$

$$= \sum_{ij} c_{ij}(\ell_{ij} + \delta) \tag{3}$$

$$\propto |\{(i,j) : \text{tuple } ij \text{ is placed in a bicluster and } e_{ij} = 1\}| \cdot \log(\frac{p}{q}) \tag{4}$$

$$+ |\{(i,j) : \text{tuple } ij \text{ is placed in a bicluster and } e_{ij} = 0\}| \cdot \log(\frac{1-p}{1-q})$$

$$= \log(P(data|\mathbf{c})) + const$$

with equality when each $r_k = N_k/M_k$. Equality (3) follows from the fact that $\sum_k N_k M_k$ is the number of clustered tuples plus the amount of overlap between the clusters, and this overlap term cancels the term $\sum_{ij} c_{ij}(\sum_k c_{ij}^k - 1)$. Step (4) follows from the choice of $\ell_{ij}$. This completes the proof of 2.3.

# 3  Automatic parameter tuning

## 3.1  Automatically tuning $r_k$

Automatically tuning $r_k$ to fit the observed biclusters allows BCMP to find differently-shaped clusters. After each message update, it updates $r_{new} = \sqrt{M_k/N_k} \cdot \sqrt{r_{old}}$.

This strategy works well in practice. It can err by finding two biclusters with shapes closer to the original choice of $r_k$ instead of a single, differently shaped cluster; to catch such errors, it is helpful to visualize the biclustering results or to check that no two clusters have mostly the same rows or columns.

## 3.2  Automatically tuning $\delta$

The offset parameter $\delta$ is tuned by maximizing the unthresholded likelihood function. BCMP is run for several values of $\delta$, and for each solution, the likelihood is computed. The solution with the greatest likelihood is kept.

## 3.3  Automatically detecting the number of clusters

When the true number of clusters is unknown, a penalty for each nonempty cluster can be used to find fewer than $K$ clusters. A term $-l_1 \max(0, N_k - 1)$ is added to the $\eta_k$ function, and $-l_1 \max(0, M_k - 1)$ is added to the $\mu_k$ function. This penalty can be interpreted as a significance threshold, as clusters with barely-positive scores might be found even in a dataset with no true clusters; or, it may be interpreted as a prior on the true number of clusters, which is $Geom(\exp(-2l_1))$ as $K \to \infty$.

Figure 1: A graphical model of the BCMP objective function.

It does not change the computational complexity to use these penalties. When computing $n_{ij}^k(1)$, there is no additional computation as the penalty $l_1$ is incurred automatically. When computing $n_{ij}^k(0)$, if $y$ is the original message before accounting for $l_1$, the new message is $\max(0, y - l_1)$.

In order to set this parameter, one strategy is: randomly permute the entries of the input matrix $(\ell_{ij})$; run BCMP on the scrambled data matrix; and choose $l_1 = \frac{1}{2} \max_k \sum_{i,j} c_{ij}^k \ell_{ij}$.

## 4   The max-sum algorithm

In the following, we explain how message passing is used to optimize the BCMP objective function in approximately linear time $\mathcal{O}(K(N + \log M)(M + \log N))$. First, we briefly explain the max-sum algorithm that we use in our optimization. Consider the following optimization:

$$\max_{X_1, \ldots, X_n} \sum_{i=1}^m f_i(x_1, \ldots, x_n) \tag{5}$$

where each variable $X_i$ has alphabet $\mathcal{X}_i$. We assign a *function node* to each function $f_i$ and a *variable node* to each variable. Messages $m_{X_i \to f_j}(x)$ and $m_{f_j \to X_i}(x)$ are vectors with length $|\mathcal{X}_i|$ defined as follows:

$$\begin{cases} m_{f_j \to X_i}(x) = \max_{X|X_i = x} \left( f_j(X_1, \ldots, X_n) + \sum_{k \neq i} m_{X_k \to f_j}(X_j) \right), \\ m_{X_i \to f_j}(x) = \sum_{k \neq j} m_{f_k \to X_i}(x). \end{cases} \tag{6}$$

When messages converge after several iterations, an optimal value of variable $X_i$ is then computed as,

$$x_i^* = \arg\max_{x \in \mathcal{X}_i} \sum_j m_{f_j \to X_i}(x). \tag{7}$$

If variables are binary (i.e., $|\mathcal{X}_i|$=2), scalar messages can be passed among nodes defined as follows,

$$\begin{cases} m_{f_j \to X_i} \triangleq m_{f_j \to X_i}(1) - m_{f_j \to X_i}(0), \\ m_{X_i \to f_j} \triangleq m_{X_i \to f_j}(1) - m_{X_i \to f_j}(0). \end{cases} \tag{8}$$

When the graphical model is a tree, the max-sum algorithm reduces to the Viterbi algorithm, which is exact. On graphical models with loops, it often obtains an approximately optimal solution in practice. It is the zero-temperature version of Belief Propagation (BP), also known as the sum-product algorithm or the cavity method. BP finds marginal probabilities in a graphical model given the joint probability distribution. To find a marginal probability, all possible configurations of the other variables must be considered, by summing instead of taking a maximum over the possible configurations. The fixed points of BP correspond to local minima of the Bethe free energy [1].

The max-sum algorithm was notably used in reference [2] for a clustering algorithm, Affinity Propagation, which inspired the proposed method.

## 5   Message updates

After solving the optimization of equation (11) in the main text, there are a few details involved in computing the actual message values. Let $n$ be the value of $N_1$ that maximizes (11), and let $y$ be the solution to equation (11).

(i) For $n_{12}^1(0)$, because the message sum is over $(i,j) \neq (1,2)$, $m_{12}^1 + t_{12}^1$ is subtracted from $s_i$. This is only significant if row 1 was included in the $\arg\max$ to (11) (i.e., if $s_1 \geq s_{(N+1-n)}$); in that case, it might no longer be optimal to include row $i$, in which case it may or may not be replaced by the $(n+1)$st largest row sum. If $l_1 = 0$, the message is

$$n_{12}^1(0) = y - \min(\max(0, m_{12}^1 + t_{12}^1), s_1 - s_{(N-n)}).$$

$s_1$ must be similarly adjusted for $n_{12}^1(1)$; however, the $\arg\max$ is not affected:

$$n_{12}^1(1) = y - \max(0, m_{12}^1 + t_{12}^1).$$

(ii) For the message $n_{12}^1(1)$, row $i$ must be included whether or not it is optimal in (11), thus it is possible that one fewer row to be included. If $s_1 < s_{(N+1-n)}$, then (i) is not relevant, and

$$n_{12}^1(1) = y + s_1 - \min(\frac{\delta}{2}(2n+1), s_{(N+1-n)})$$

(This also assumes that $l_{12} \geq 0$).

(iii) For calculating $n_{12}^1(0)$, if the initial penalty $l_1$ is used (see Supplementary Note 3.3), it might be optimal to exclude all rows and set every $c_{12}^1 = 0$. If $y'$ is the message value for $l_1 = 0$, then accounting for $l_1$, the true value is $\max(0, y' - l_1)$. For $n_{12}^1(1)$, the $\arg\max$ is unaffected, and the message value is $y' - l_1$.

## 6 Expectation-Maximization for the binary and Gaussian models

This section describes how expectation-maximization (EM) was applied to the binary and Gaussian models used in our simulations (see Section 2.4) .

In the case of a binary model with parameters $p$ and $q$, the prior on $p$ is $Unif(0,1)$. If $n$ tuples were placed in a cluster, and $m$ out of those $n$ had edges, the posterior distribution for $p$ is $Beta(1+m, 1+n-m)$, and the posterior for $q$ is computed similarly. The posteriors are independent of each other because the edge weights are assumed to be drawn independently (given the cluster assignments). There are only two integrals to be computed:

$$l_{ij} = \begin{cases} \int_{[0,1]^2} \log \frac{p}{q} dP(p)dP(q) & e_{ij} = 1 \\ \int_{[0,1]^2} \log \frac{1-p}{1-q} dP(p)dP(q) & e_{ij} = 0 \end{cases} \tag{9}$$

Here we use $P(p)$ and $P(q)$ to denote the posterior CDF of these parameters. These integrals can be computed numerically by sampling from the posterior distributions.

In the case of a Gaussian model with two different means and variances, we use a normal-inverse gamma distribution on the mean and variance. Assume that $e_{ij}|c_{ij} = a$ is normally distributed with mean $\mu_a$ and variance $\sigma_a^2$. Let the prior distribution both within clusters and outside of them be $(\mu_a, \sigma_a^2) \sim NIG(\mu, \lambda, \alpha, \beta)$, that is,

$$\sigma_a^2 \sim \Gamma(\alpha, \beta), \tag{10}$$
$$\mu_a|\sigma_a^2 \sim N(\mu, \sigma_a^2/\lambda).$$

Let $\hat{\mathbf{c}} = (\hat{c_{ij}})$ be the current estimate of the cluster memberships. Let $A \triangleq \{(i,j) : \hat{c_{ij}} = 1\}$ and $B \triangleq \{(i,j) : \hat{c_{ij}} = 0\}$; let $\hat{\mu_A}, \hat{\sigma}_A^2, \hat{\mu_B}, \hat{\sigma}_B^2$ be the sample means and variances of the edge weights over the estimates $\hat{A}, \hat{B}$. Then the posterior distributions of the model parameters are

$$(\mu_1, \sigma_1^2)|\hat{A} \sim NIG(\frac{\mu\lambda^{-1} + |A|\hat{\mu_A}}{\lambda^{-1} + |\hat{A}|}, \lambda^{-1} + |\hat{A}|, \alpha + \frac{|\hat{A}|}{2}, \beta + \frac{1}{2}\hat{\sigma}_A^2) \tag{11}$$

$$(\mu_0, \sigma_0^2)|\hat{B} \sim NIG(\frac{\mu\lambda^{-1} + |B|\hat{\mu_B}}{\lambda^{-1} + |\hat{B}|}, \lambda^{-1} + |\hat{B}|, \alpha + \frac{|\hat{B}|}{2}, \beta + \frac{1}{2}\hat{\sigma}_B^2)$$

Let $\phi$ be the standard Normal density function. The expression for the new input matrix is:

$$l_{ij} = \int_{\mathbb{R}^2} \log \sigma_1^{-1} \phi(\frac{e_{ij} - \mu_1}{\sigma_1}) dP(\mu_1, \sigma_1) - \int_{\mathbb{R}^2} \log \sigma_0^{-1} \phi(\frac{e_{ij} - \mu_0}{\sigma_0}) dP(\mu_0, \sigma_0) \qquad (12)$$

This computation can be performed efficiently by drawing samples from the two distributions and reusing them for each of the $NM$ integrations. Typically, in our simulations, approximately three iterations of EM were needed for convergence.

## 7    Maximum Modularity

When a statistical model for the data is not available, a natural heuristic input matrix, replacing $(l_{ij})$, is the *modularity matrix* [3] with entries

$$M_{ij} = e_{ij} - \frac{d_i d_j}{2m},$$

where $d_i, d_j$ are the degrees of nodes $i$ and $j$, and $m = \frac{1}{2} \sum_{i=1}^{N} d_i$ is the total weight of the graph. The term $\frac{d_i d_j}{2m}$ is interpreted as the expected weight of $e_{ij}$ under a null model that accounts for the degree of each node; it satisfies the condition $\sum_{i=1}^{N} E(e_{ij}) = d_j$ for each column $j$ and likewise for each row $i$, where $E()$ indicates the expectation. This matrix was originally used for graph partitioning in the case of a binary, non-bipartite graph; however, it can be extended to the overlapping, weighted and bipartite cases.

In the non-partitioning case, typically most nodes will be assigned to some cluster using the maximum modularity method, for two reasons. First, if $(U_1, V_1)$ is a bicluster of modularity $x$, then there will be a "reflected" cluster $(U_1^c, V_1^c)$ that also has modularity $x$ even though it might be no denser, in terms of actual edge weights, than the network mean. Second, because the row and column sums of $M$ are all zero, there is a high probability that a node will be incorrectly assigned to any given cluster. If $V_1$ is a collection of columns and $|V_1| << M$, then $E(\sum_{j \in V_1} M_{ij}) \approx 0$, which implies that node $i$ will be assigned to the cluster erroneously with probability approaching $1/2$. To avoid finding spurious clusters and erroneously assigning extra nodes to clusters, therefore, we recommend using a cluster-size penalty that is larger than the offset. This value can be chosen by drawing a number of random "biclusters", computing their respective modularities, and choosing a value such that all but a fraction of them have negative shifted modularity.

## 8    Gene expression data

We applied BCMP, ISA and LAS to three DREAM5 gene expression datasets: *In Silico*, *E. coli*, and *S. cerevisiae* [4]. We binarized these datasets, placing ones where the gene expression level was at least two standard deviations from its mean and zeroes elsewhere. We evaluated the three biclustering algorithms in terms of the total size and average density of the reported clusters; when reported clusters overlapped, these regions were not double-counted. We used BCMP to fit the stochastic block model, using different initial parameter settings for EM (see section 2.4 and supplementary section 6) and $K = 10$ (this is the number of clusters reported by LAS by default). ISA was run with different threshold-parameter settings, keeping the first ten clusters reported. LAS had no parameters to set. We found that the density of the clusters reported by LAS was diluted by a few large, low-density clusters. ISA and BCMP had similar results for the *In Silico* and *E. coli* datasets, and BCMP reported denser clusters on the *S. cerevisiae* datasets (see Supplementary Figure 2).

## 9    Document classification example

Biclustering has been used for document clustering [5]. Here we illustrate of how BCMP might be applied to such a problem.

Let $D = d_1, ..., d_N$ be a collection of documents containing the words $W = w_1, ..., w_M$, and let $(D_1, W_1), ..., (D_K, W_K)$ be a hidden set of topics (each word and document can belong to any

Figure 2: Total size and average density of biclusters reported by BCMP, ISA and LAS for three gene expression datasets. a) *In Silico*. b) *E. coli*. c) *S. cerevesiae*.

number of topics). Document $d_i$ contains $n_{ij}$ instances of word $w_j$ out of $n_i = \sum_j n_{ij}$ total words. Assume that $n_i$ is large enough to use a Poisson model: $n_{ij} \sim Poisson(\lambda_{ij})$ where $\lambda_{ij} = n_i r_j$ if document $d_i$ concerns a topic relating to word $w_j$ and $\lambda_{ij} = n_i s_j$ otherwise. Assume that $K$ is given.

The problem is to recover the topics. It can be solved by estimating the parameters $r_{ij}, s_{ij}$ and biclustering the bipartite graph $(D, W, (n_{ij}))$.

We can obtain initial estimates for $r_j$ and $s_j$ using k-means on $n_{1j}, ..., n_{Nj}$. This is effectively a clustering problem along one axis with two clusters:

$$\max_{r_j, s_j, c_{1j}, ..., c_{Nj}} P(n_{1j}, ..., n_{Nj}|r_j, s_j, c_{1j}, ..., c_{Nj})$$

where $c_{ij}$ is the indicator variable for document $i$ concerning some topic relating to word $j$. Words with no evidence for $r_j \neq s_j$ at some significance threshold can be discarded. After performing biclustering, these estimates can be improved using the newly estimated cluster memberships.

After estimating $r_j$ and $s_j$, the likelihood ratios are

$$\frac{p_{ij}}{q_{ij}} = \frac{P(n_{ij}|\lambda_{ij} = n_i r_j)}{P(n_{ij}|\lambda_{ij} = n_i s_j)} = \frac{(n_i r_j)^{n_{ij}} e^{-n_i r_j}/n_{ij}!}{(n_i s_j)^{n_{ij}} e^{-n_i s_j}/n_{ij}!}$$

and the input matrix for BCMP is

$$\ell_{ij} = \max(0, \log \frac{p_{ij}}{q_{ij}} + \delta) = \max(0, n_{ij} \log(\frac{r_j}{s_j}) + n_i(s_j - r_j) + \delta). \tag{13}$$

EM can be used to iteratively update the input matrix and improve the parameter estimates as discussed in Section 2.4.

Figure 3: Running time for BCMP. In Proposition 3, we showed that the complexity of a round of message updates for BCMP is nearly linear, i.e., $\mathcal{O}(K(N + \log M)(M + \log N))$. This figure illustrates the running time of BCMP on a personal computer, with $N = M$.

## 10 Pseudocodes for BCMP

The pseudocode is presented as three functions. The first calls the two message-update functions, computes clusters from messages, and checks for convergence.

The second function computes messages from the $\mu_k$ and $\eta_k$ function nodes. The steps for computing these messages are explained here briefly; see Section 2.3 and Supplementary Note 5 for a derivation of the message update rules. An optimization is performed under two constraints: either tuple $(i, j)$ must be assigned to cluster $k$ ($c_{ij}^k = 1$), or it must not be. While the optimization is over $NM$ variables, only the number of rows or columns (for $\eta$ and $\mu$ respectively) is penalized; thus, the optimum is achieved by including every tuple $(i, j)$ with positive outgoing message for some number of rows $i$ (or columns $j$). When tuple $(i, j)$ is constrained to be in the cluster, node $i$ must also be in the cluster; however, node $i$ may still be in the cluster even when tuple $(i, j)$ is constrained not to be. The steps are, first, to compute the sum of the positive incoming messages for each node $i = 1, ..., N$ and sort them. Second, to find the unconstrained $\arg\max$ by comparing the sorted list of message sums with the marginal penalty for including another node in the cluster. Three indices are computed: $t_0, t_1, t_2$, which are the numbers of nodes included in the cluster under various constraints. $t_1$ is the index for the unconstrained maximum. When an extra node, not included in the unconstrained maximum, is constrained to be included, $t_2 \leq t_1$ is the optimal number of additional nodes to also include. $t_0 \in \{t_1, t_1 + 1\}$ is the optimal number when a node drops out of the sum (owing to one of the tuples being excluded). The third step is to compute the constrained $\arg\max$ for each tuple; one of the constraints gives the unconstrained maximum, and the other gives an optimization that is solved by including either $t_0$ or $t_2$ nodes from the sorted list of message sums.

The third function computes messages from the $\tau_{ij}$ function nodes. First, it finds the largest and second largest messages $msg_k$, and the $\arg\max$ $k_1$; the second largest message value is needed to compute the message for $k = k_1$. Then, it computes the message values explicitly.

---
**Algorithm 1** Biclustering Using Message Passing
___

    **function** BCMP$((\ell_{ij}), K, l_0, r)$

**Require:** $N \times M$ dataset $(l_{ij})$; maximum number of clusters to find $K$; penalty per nonempty cluster $l_1$.

        $(t_{ij}^k) \leftarrow 0$

        $(n_{ij}^k) \leftarrow random$                                                     ▷ small random values

        $(m_{ij}^k) \leftarrow 0$

        **for** $rep = 1, ..., repmax$ **do**

            **for** $i = 1, ..., N$ **do**

                **for** $j = 1, ..., M$ **do**

                    $(t_{ij}^1, ..., t_{ij}^k) \leftarrow (t_{ij}^1, ..., t_{ij}^k) \cdot \lambda_1 + tuple\_update(n_{ij}^1 + m_{ij}^1, ..., n_{ij}^k + m_{ij}^k) \cdot (1 - \lambda_1)$

                **end for**

            **end for**

            **for** $k = 1, ..., K$ **do**

                $(n_{ij}^1, ..., n_{ij}^k) \leftarrow (n_{ij}^1, ..., n_{ij}^k) \cdot \lambda + pen\_update((t_{ij}^k) + (n_{ij}^k), l_0 r_k) \cdot (1 - \lambda)$   ▷ $\lambda$ is a damping factor; a good value is $\lambda = 1/2$. The input to the penalty message update function is a $NxM$ or $MxN$ matrix, and the penalty coefficient.

            **end for**

            **for** $k = 1, ..., K$ **do**

                $(m_{ij}^1, ..., m_{ij}^k) \leftarrow (m_{ij}^1, ..., m_{ij}^k) \cdot \lambda + pen\_update((t_{ij}^k)^T + (m_{ij}^k)^T, l_0/r_k) \cdot (1 - \lambda)$

            **end for**

            ▷ Compute biclusters from current messages in order to update parameters and check for convergence.

            $(c_{ij}^k) \leftarrow (n_{ij}^k + m_{ij}^k + t_{ij}^k > 0)$                   ▷ $N \times M \times K$ boolean array

            $(ind_{ij}) \leftarrow (\sum_k c_{ij}^k > 0)$                          ▷ $N \times M$ boolean array

            $score \leftarrow \sum_{ij} ind_{ij}(l_{ij} - 2l_0)$

            **if** $score > oldscore$ **then**

                $oldscore \leftarrow score$

            **else**

                break

            **end if**

        **end for**

        **return** $(t_{ij}^k) + (n_{ij}^k) + (m_{ij}^k)$

**end function**
___

---

**Algorithm 2** Message update function for $\eta$ and $\mu$

---

   **function** PEN_UPDATE($(msg_{ij}), a, b$)

**Require:** $(msg_{ij})$ the $N \times M$ matrix of messages from the variable nodes; $a$ the penalty coefficient; $b$ the initial penalty per nonempty cluster.

      $(sums_1, ..., sums_N) \leftarrow (\sum_j \max(0, msg_{1i}), ..., \sum_j \max(0, msg_{Ni}))$            $\triangleright$ Step one

      sort ($sums$) and store the permutation as $\sigma : 1, ..., N \to 1, ..., N$

      $(t_1, ..., T_N) \leftarrow a \cdot (1, 3, 5, ..., 2N - 1)$         $\triangleright T_i = ai^2 - a(i-1)^2$ the marginal penalty

                                                                                              $\triangleright$ Step two

      $t_1 \leftarrow \min(\{n : msg_{\sigma(n)} - T_n < 0\}) - 1$         $\triangleright$ or $N$ if this set is empty

      $t_2 \leftarrow \min(\{n : sums_{\sigma(n)} - T_{n+1} < 0\}) - 1$       $\triangleright$ or $N$ if this set is empty

      **if** $0 < t_1 < N$ **then**

          **if** $sums_{\sigma(n)} \geq T_{t_1}$ **then**

              $t_0 \leftarrow t_1 + 1$

          **else**

              $t_0 \leftarrow t_1$

          **end if**

      **else if** $t_1 = 0$ **then**

          **for** $i = 1, ..., N$ **do**

              $newmsg_{i1}, ..., newmsg_{iM} \leftarrow b - a + sums_i - (\max(0, msg_{1i}), ..., \max(0, msg_{Ni}))$

          **end for**

        **return** $(newmsg_{ij})$

      **else**

          $t_0 \leftarrow t_1$

      **end if**

      $sum\_to\_t_0 \leftarrow \sum_{i=1}^{t_0} sums_{\sigma(i)}$

      $sum\_to\_t_1 \leftarrow \sum_{i=1}^{t_1} sums_{\sigma(i)}$

      $sum\_to\_t_2 \leftarrow \sum_{i=1}^{t_2} sums_{\sigma(i)}$

      $unconstrained\_max \leftarrow b + sum\_to\_t_1 - a \cdot t_1^2$

                                                                                              $\triangleright$ Step three

      **for** $i = 1, ..., t_1$ **do**

          **for** $j = 1, ..., M$ **do**

              $newmsg_{\sigma(i),j} \leftarrow unconstrained\_max - \max(0, msg_{\sigma(i),j}) - \max(b + \max(0, sum\_to\_t_1 - \max(0, msg(\sigma(i), j)) - a \cdot t_1^2, sum\_to\_t_0 - sums_{\sigma(i)} - a \cdot (t_0 - 1)^2)$

          **end for**

      **end for**

      $unconstrained\_max \leftarrow \max(0, unconstrained\_max)$

      **for** $i = t_1 + 1, ..., N$ **do**

          **for** $j = 1, ..., M$ **do**

              $newmsg_{\sigma(i),j} \leftarrow b + sum\_to\_t_2 + sums_{\sigma(i)} - \max(0, msg_{\sigma(i),j}) - a \cdot (t_2 + 1)^2 - unconstrained\_max$

          **end for**

      **end for**

        **return** $(newmsg)$

   **end function**

---

---

**Algorithm 3** Message update function for $\tau$

---

**function** TUPLE_UPDATE$((msg_k), \delta, \ell_{ij})$
    **for** $k = 1, ..., K$ **do**
        **if** $w1 <= msg_k$ **then**
            $k_1 \leftarrow k$
            $w_2 \leftarrow w_1$
            $w_1 \leftarrow msg_k$
        **else if** $w_2 <= msg_k$ **then**
            $w_2 = msg_k$
        **end if**
    **end for**
    **for** $k = 1, ..., K$ **do**
        **if** $k \neq k_1$ **then**
            $newmsg_k = \ell_{ij} - \max(0, \delta + msg_k) - [\max(0, \ell_{ij}) + w_1 - \max(0, \delta + msg_k) - \max(0, 2\delta + w_1)]$
        **else**
            $newmsg_k = \ell_{ij} - \max(0, \delta + msg_k) - [\max(0, \ell_{ij}) + w_2 - \max(0, \delta + msg_k) - \max(0, \delta + w_2)]$
        **end if**
    **end for**
    **return** $(newmsg)$
**end function**

---