[Reviews · NeurIPS 2014]

Submitted by Assigned_Reviewer_43

In this paper the authors propose a novel bi-clustering approach based on a message passing algorithm. The motivation is that most current bi-clustering techniques overcome the computational difficulty of the problem by performing greedy local optimisations. In this paper, the authors propose to overcome this by defining a global likelihood function (eq 1) and then maximising an approximation to this function via message passing.

Quality: this is a high quality paper. The motivation is strong, and the solution looks very good. I think that this is certainly of a high enough standard for NIPS. I don't really have any major criticism of this paper. My only minor gripe would be that the authors make no comparison of computational run-times which, given the computational difficulty of the problem, are probably quite relevant.

Clarity: very clearly written, easy to understand.

Originality and significance: as the authors argue, bi-clustering is a common problem and the proposed solution looks very strong. As such I think this paper is highly significant and original.
Summary: This paper describes an algorithm for bi-clustering (simultaneously clustering rows and columns of a matrix) that uses message passing to optimise an approximation to a global likelihood function. The message passing algorithm performs very well in the comparisons.

Submitted by Assigned_Reviewer_46

This paper is about biclustering. It presents a cost model and algorithm for biclustering that ensures that it recovers biclusters generated by a particular (natural) stochastic model. The main idea is to build a simple cost function that can be optimized using a message passing system. In order to this, some care is needed in how the cost function is built, so that the overall function can be written as a max sum.

The results provided are reasonable: the algorithm is good at identifying biclusters that might overlap, and also works well upto some level of noise in the data.

One thing that is missing is a discussion of runtime: is it the case that this algorithm converges faster than prior methods ? does it spend more iterations, but cheaper ones ? and so on ? A message passing architecture has very lightweight iterations, but maybe it takes longer to converge ?

Summary: A decent paper on an important problem. I like the link to learning a particular statistical model.

Submitted by Assigned_Reviewer_47

This paper proposes a biclustering algorithm that has some advantages over related works. It is able to deal with overlapping clusters and the clustering problem is addressed by the maximization of a well-defined global objective function.
I find the idea interesting and the paper technically sound and well written. Despite the description of related works is kind of short, it is enough for positioning the paper in the right context and to understand the novelty of the proposals.
While the theoretical formulation presented in Section 2 seems to be correct, it would be very beneficial to have at the end of this section the pseudo-code of the proposed BCMP method. This would help to wrap up the results presented in this section and to make clearer the steps of the proposed algorithm.
In the experimental section, the proposed method is compared with to related methods on simulated data with different degrees of noise and overlapping between clusters. Despite these results illustrate the suitability of the proposed algorithm; it would be nice to see its performance on real datasets. In the introduction of the paper, authors claimed that this kind of algorithms can be used in different kind of applications like clustering of microarray data, tumor prediction and document classification. However, no real data is used to test the performance (and compare with existing methods) of the proposed clustering algorithm.
A final minor comment is regarding the abstract. In the current version is too long, at some point it looks like an introductory section. I think it can be reduced to half of the size.
Summary: Interesting idea and technically sound. It can be improved by evaluating the performance of the proposed method on real datasets and comparing it with related methods.
Author Feedback
Author rebuttal: We thank the reviewers for their constructive and thoughtful comments. Below, we make some common responses before addressing reviewer-specific comments point-by-point.

Common Response 1 (CR1): application to gene regulatory data

We have now applied our method to three binarized DREAM5 gene expression datasets: in silico (1,643 genes and 487 treatments), E. coli (4,511 by 487) and S. cerevisiae (5,950 by 321). We use Expectation-Maximization (EM, see CR2) to learn the model parameters (p and q of Definition 2). In all cases, individual clusters inferred by our method have comparable quality, in terms of density for given size, to high-density clusters inferred by ISA and LAS. After aggregating the inferred clusters for each method, BCMP outperforms ISA and LAS, both of which were more liable to report low-density clusters. We conclude that BCMP is able to perform well, in terms of this simple objective, in real datasets even when (1) the exact generative model is not known, and (2) the parameters of the simplified model need to be learned. In the revised draft, we will provide the figures.

CR2: learning model parameters

Although there was no particular comment on this topic, we decided that it was important to consider the situation that the model parameters (e.g., p and q of Definition 2) are unknown. This is likely the case in most applications; fortunately, we can use EM to learn these parameters. In the M step, we use BCMP to infer clusters. In the E step, we use inferred clusters to compute the expected log-likelihood matrix for the next M-step. In the binary case, we use a uniform prior on p and q, which leads to Beta posteriors after observing Binomial emissions. In the Gaussian case, we use an uninformative (large variance) normal inverse gamma prior distribution on the means and variances.

We re-ran simulations using EM to learn model parameters. For lower noise levels, EM-based BCMP performs equally well as when true parameters are given, while in higher noise, it performs marginally worse. The qualitative conclusions from our simulations are not altered.

In the revised text, we will further explain parameter selection and learning.

CR3: running time

In Theorem 1, we show that complexity of BCMP is nearly linear if the number of iterations performed is constant. We have now added two figures in response to the reviewers' questions. The first illustrates the running time of different methods. To give an example, over a 500 by 2,000 matrix, one run of our method takes approximately 250 seconds on an ordinary laptop; as implemented, our method is slower by a constant factor than ISA and LAS. The second figure shows that the number of iterations needed for convergence does not increase with the size of the dataset, at least for the particular generative model used (binary, with q=.3, p=.7, and three disjoint planted clusters). In fact, the algorithm is able to converge more rapidly for larger datasets due to the strength of the signal.

Reviewer 1 comments

Comment: The motivation is strong, and the solution looks very good. I think that this is certainly of a high enough standard for NIPS.

Response: We thank the reviewer for positive evaluation of our paper.

Comment: My only minor gripe would be that the authors make no comparison of computational run-times which, given the computational difficulty of the problem, are probably quite relevant.

Response: Please see CR3.

Reviewer 2 comments

Comment: The results provided are reasonable: the algorithm is good at identifying biclusters that might overlap, and also works well upto some level of noise in the data.

Response: We thank the reviewer for positive evaluation of our paper.

Comment: One thing that is missing is a discussion of runtime: is it the case that this algorithm converges faster than prior methods? does it spend more iterations, but cheaper ones ? and so on ? A message passing architecture has very lightweight iterations, but maybe it takes longer to converge?

Response: Thanks for the comment. Please see our response in CR3.

Reviewer 3 comments

Comment: I find the idea interesting and the paper technically sound and well written.

Response: We thank the reviewer for positive evaluation of our paper.

Comment: While the theoretical formulation presented in Section 2 seems to be correct, it would be very beneficial to have at the end of this section the pseudo-code of the proposed BCMP method.

Response: We will include pseudo-code in our revised draft.

Comment: In the experimental section, the proposed method is compared with to related methods on simulated data with different degrees of noise and overlapping between clusters. Despite these results illustrate the suitability of the proposed algorithm; it would be nice to see its performance on real datasets. In the introduction of the paper, authors claimed that this kind of algorithms can be used in different kind of applications like clustering of microarray data, tumor prediction and document classification. However, no real data is used to test the performance (and compare with existing methods) of the proposed clustering algorithm.

Response: Thanks for the comment. Now we have included a real data application to our paper. Please see CR1.

Comment: A final minor comment is regarding the abstract. In the current version is too long, at some point it looks like an introductory section. I think it can be reduced to half of the size.

Response: We will shorten the abstract in the final draft.
Response: We will do our best to shorten it.